# Training Specificity in Trail Running: A Single-Arm Trial on the Influence of Weighted Vest on Power and Kinematics in Trained Trail Runners

**DOI:** 10.3390/s23146411

**Published:** 2023-07-14

**Authors:** Antonio Cartón-Llorente, Alberto Rubio-Peirotén, Silvia Cardiel-Sánchez, Luis E. Roche-Seruendo, Diego Jaén-Carrillo

**Affiliations:** 1Universidad San Jorge, 50830 Zaragoza, Spain; acarton@usj.es (A.C.-L.); scardiels@usj.es (S.C.-S.); leroche@usj.es (L.E.R.-S.); diego.jaen@uibk.ac.at (D.J.-C.); 2Department of Sport Science, University of Innsbruck, 6020 Innsbruck, Austria

**Keywords:** endurance, footpod, running power, wearable resistance

## Abstract

Participants in trail running races must carry their equipment throughout the race. This additional load modifies running biomechanics. Novel running powermeters allow further analyses of key running metrics. This study aims to determine the acute effects of running with extra weights on running power generation and running kinematics at submaximal speed. Fifteen male amateur trail runners completed three treadmill running sessions with a weighted vest of 0-, 5-, or 10% of their body mass (BM), at 8, 10, 12, and 14 km·h^−1^. Mean power output (MPO), leg spring stiffness (LSS), ground contact time (GCT), flight time (FT), step frequency (SF), step length (SL), vertical oscillation (VO), and duty factor (DF) were estimated with the Stryd wearable system. The one-way ANOVA revealed higher GCT and MPO and lower DF, VO, and FT for the +10% BM compared to the two other conditions (*p* < 0.001) for the running speeds evaluated (ES: 0.2–7.0). After post-hoc testing, LSS resulted to be higher for +5% BM than for the +10% and +0% BM conditions (ES: 0.2 and 0.4). Running with lighter loads (i.e., +5% BM) takes the principle of specificity in trail running one step further, enhancing running power generation and LSS.

## 1. Introduction

Unlike most road races, trail and ultra-trail runners need to carry their technical equipment (i.e., headlamps, spare batteries, first aid kit, etc.) from the beginning to the end of the race. This adds an additional weight to the athletes’ vest that can be up to 3–4 kg extra. Regarding this, it has been reported that running with additional external loads modifies running biomechanics [1]. Previous studies found that an extra load (5–30% body mass) increases (+4–7%) ground contact time (GCT), vertical oscillation (VO) (+12–20.5%) [1,2,3], and leg spring stiffness (LSS) (+7–14%) [1,2,4]; however, controversial results have been reported for step length (SL) [1]. Worth noting, no study included duty factor (DF) or running power in their kinematic analyses. DF refers to the ratio of GCT and total stride time and provides a deeper insight into the overall running pattern than when these spatiotemporal parameters are considered independently [5].

For its part, mechanical power output (MPO) is the rate at which mechanical work is performed, being a key variable in endurance sports. Running power has been related to the metabolic cost of running [6,7], and it has been proposed as a predictor of running performance [8]. Its main advantage over other load indicators is that running power assesses the current workload that the athlete is developing regardless of likely external conditions (i.e., wind speed, slope steepness, terrain type, or additional weight). In fact, power output has been shown to be more sensitive to small changes in exercise intensity than other commonly used internal (i.e., heart rate) and external (i.e., speed) workload indicators [9]. In this regard, the Stryd running powermeter showed the highest concurrent validity with metabolic power measurements (i.e., VO2máx) (r ≥ 0.911, SEE = 7.3%), and the most repeatable device for running power estimation (SEM ≤ 12.5 W, CV ≤ 4.3%, ICC ≥ 0.980) [6]. Additionally, it has been considered the best wearable tool to analyze running power given its high values of repeatability [6]. As a result, the use of this foot pod has been spread among athletes and coaches of both road and trail running, generating a parallel interest among researchers in sports science.

The emergence of running powermeters, and their friendly use and interpretation of the running power metric, have contributed to their spread amongst athletes and coaches, who are now considering running power as a load indicator. By analyzing the effects of overloaded running on running performance parameters, the optimal additional load to optimize the generation of running power without jeopardizing running performance might be determined, which is one of the objectives of wearable endurance training [1]. In this regard, the use of external additional loads (i.e., weighted vests) during running is a practical example of how the concept of training specificity can be applied in trail running. This has been shown to elicit adaptations in the neuromuscular behavior of the legs to reuse elastic energy and improve running economy [10] by optimizing the behavior of the stretch-shortening cycle and lower-limb stiffness. Theoretically, higher “leg-spring” stiffness maximizes potential elastic energy return, improving running economy [11]. However, the influence of mechanical stiffness on running performance is specific to the individual, and the assumption is still controversial.

Although several studies have evaluated the acute and longitudinal effects of running and training with additional load, this was mainly analyzed in sprint running [12,13]. A recent study from Cerezuela-Espejo and colleagues measured MPO when running with additional load (i.e., +2.5 and +5 kg) with the aim of determining the test–retest reliability of the Stryd running powermeter [6]. Unfortunately, the likely influence of additional load on running power was not considered. Therefore, the present study attempts to determine the acute effects of running with additional load (i.e., +5 and +10% of body mass) on running power generation and running kinematics at submaximal speeds (i.e., 8, 10, 12, and 14 km/h). We hypothesized that running with +5% BM would optimize running power generation without impairing running kinematics, whereas running with +10% BM will cause major changes in running technique to be able to produce the required amount of power.

## 2. Materials and Methods

### 2.1. Participants

A group of fifteen male amateur trail runners (age: 37 ± 6 years; height: 1.76 ± 0.04 m; body mass: 72.6 ± 5 kg) voluntarily participated in this study. All participants met the inclusion criteria: (i) men older than 18 years old, (ii) at least 2 years of experience in trail running, (iii) not suffer any lower-limb injury in the last 6 months before the data collection that would make running training impossible for more than 2 weeks (iv), no cardiorespiratory or metabolic abnormality, and (v) not be taking any type of ergogenic aid that could disturb the results of the study. After receiving detailed information on the objectives and procedures of the study, each subject signed an informed consent form prior to participation, which complied with the ethical standards of the World Medical Association’s Declaration of Helsinki (2013). It was made clear that the participants were free to leave the study if they saw fit. The study was approved by the Ethics Committee of San Jorge University (Zaragoza, Spain).

Sample size power calculation was executed using G*POWER 3.1.9.7 (University of Dusseldorf, Dusseldorf, Germany). The following structure was used based on the analysis: F test-ANOVA: repeated measures, within-interaction; A priori. Effect size f = 0.5; α error prob = 0.05; power (1-β error prob = 0.95; The number of groups = 1n and the number of measurements = 3n. The result showed a suitable total sample size of 12 athletes for actual high power (95.23%).

### 2.2. Procedures

This study was conducted in four sessions (Figure 1). During the first session, a consent form was obtained from every participant; moreover, anthropometric measurements were collected, and the participants became familiar with the running protocol. During days 2, 3, and 4, the running protocol was carried out with the corresponding additional load (i.e., 0, 5, and 10% additional body weight). To avoid any bias, the additional load with which each had to be run was randomized. Prior to all testing, subjects refrained from severe physical activity for at least 72 h, and all tests were performed at least 3 h after eating. The tests were performed with the subjects wearing their usual running shoes to measure their typical performance.

The participants performed an incremental running protocol on a motorized treadmill (HPCosmos 20, Nußdorf, Germany) at an initial speed of 8 km·h^−1^ for 3 min. Then, the speed was increased by 1 km·h^−1^ every minute until volitional exhaustion. The slope was maintained at 1% over the entire protocol to simulate external conditions of air resistance [14]. No feedback was given to participants during data collection and subsequent analysis was performed by a different researcher for all measurements and conditions at the same time.

### 2.3. Materials and Testing

For descriptive purposes, body height (cm), body mass (kg), and body fat (% body weight) were determined using a precision stadiometer and a weighing scale (SECA 222 and 634, respectively, SECA Corp., Hamburg, Germany). All measurements were taken with the participants wearing underwear. Body mass index (BMI) was calculated from the subjects’ body mass and height (kg·m^−2^) (Table 1).

The spatiotemporal variables of GCT, flight time (FT), DF, SF, VO, and SL; MPO (in W), power output normalized to body mass (nMPO), and LSS was estimated with the Stryd running powermeter (Stryd powermeter, Stryd Inc., Boulder, CO, USA).

Stryd is a carbon fiber-reinforced foot pod based on a 6-axis inertial motion sensor (3-axis gyroscope, 3-axis accelerometer), which has been shown reliable (CV ≤ 3%, ICC ≥ 0.95) [6] and valid (Pearson r: 0.82 to 0.94, compared to a reference system) for running kinematic analyses [15]. Data from Stryd™ were obtained from its website (www.stryd.com/powercenter/analysis, accessed on 16 June 2023) into the .csv file. Those files were imported into Excel^®^ (2016, Microsoft, Inc., Redmond, WA, USA) and further analyzed in SPSS (described below).

The additional load (i.e., +5% and +10% of body weight) was added by using a weighted vest. This has a series of pockets on the front and back where 300 g bags can be added, allowing accurate control of the additional load and an even distribution of the extra load.

### 2.4. Statistical Analysis

Descriptive statistics are presented as means ± standard deviations (SD) for all variables. Comparisons between conditions (i.e., +0% BM vs. +5% BM, +0% BM vs. +10% BM, and +5% BM vs. +10% BM) are presented as mean difference ± SD. The middle 30 s of each speed were included in the analyses to avoid errors derived from adaptation to the different speeds. The Kolmogorov–Smirnov test was conducted to confirm data distribution normality and Levene’s test for equality of variances. A separate one-way analysis of variance (ANOVA) was used to identify differences between conditions regarding the extra weight for the different running speeds (i.e., 8, 10, 12, and 14 km·h^−1^). Finally, Gabriel or Games-Howell post-hoc analyses were also conducted when appropriate to determine significant differences between conditions. Effect sizes for all pairwise comparisons were also calculated using Cohen’s d, with 95% confidence intervals. Cohen’s d were classified as follows: small (0.00 < d < 0.49), medium (0.50 < d < 0.79), and large effects (d > 0.8) [16]. Data analysis was performed using SPSS (version 28, SPSS Inc., Chicago, IL, USA).

## 3. Results

Table 2 shows the acute responses of the running kinematic and power-related variables under additional loads and body mass conditions and the magnitude and direction of the differences between the three conditions (i.e., 0%, +5%, and +10%).

The one-way ANOVA showed significantly higher GCT and MPO and lower DF, VO, and FT for the +10% BM compared to the two other conditions (*p* < 0.001) for the running speeds here evaluated, with effect sizes ranging from small to large (0.2–7.0) and SL showing small differences at 8 and 10 km·h^−1^. The comparison between +0% BM and +5% BM showed significant differences in LSS at all running speeds (*p* < 0.001) in favor of +5% BM, with small sizes (0.1–0.3). Moreover, the differences between the rest of the variables for these two conditions were not consistent among the analyzed speeds, with their effect sizes being small (≤0.3).

After post-hoc testing, LSS resulted to be higher for +5% BM than for the +10% BM condition at 10, 12, and 14 km·h^−1^ (effect sizes between 0.2 and 0.4), whereas SF was lower for +10% BM than for the rest of the running conditions at 8 km·h^−1^.

The acute responses of running kinematic and power-related variables to the three different conditions and speeds are shown in Figure 2.

## 4. Discussion

The present study aimed to assess the acute effects of running with additional load (i.e., +5 and +10% additional body weight) on power output and running kinematics of trained trail runners at submaximal speeds (i.e., between 8 and 14 km·h^−1^). The main findings of this work were that running with an additional load of +10% body mass reduced flight time (ES: 0.5 to 0.9) and increased DF (ES: 0.7 to 1.1) and GCT (ES: 0.4 to 0.9) in level running at 8, 10, 12, and 14 km·h^−1^, while running with an additional load of 5% body mass did not produce substantial alterations in running kinematics. Although MPO increased linearly with additional loading (ES: 2.7 to 7.0) at all velocities analyzed, running LSS remained unchanged in the +10% BM condition and subtly increased in the +5% BM condition (ES: 0.2 to 0.3). Therefore, running with lighter loads (i.e., 5% BM) seems a more appropriate strategy to increase power generation and stimulate lower extremity elastic energy reuse.

The current work found that an additional load of +10% BM significantly increases GCT and DF for all the speeds here evaluated in comparison to running with no additional load (i.e., 0% BM). These findings are supported by a previous study using similar speeds and comparing 0% BM and +10% BM conditions [2]. Although no previous study analyzing the likely influence of additional loads on running considered DF in trail or distance runners, an increase in DF alongside the increase in GCT can be expected in all BM conditions (i.e., +5% BM and +10% BM) for all the speeds analyzed as reported in previous sprint running studies [17,18] given the longer time spent on the ground. Moreover, it has also been found that such an additional load of +10% BM significantly decreases FT, SF, and VO when running at 8 km·h^−1^. Similarly, FT and VO were significantly affected by the additional load of +10% BM at 10, 12, and 14 km/h reducing their values. Regarding FT, previous research on sprint running [17,18] reported significantly reduced values when running under loaded conditions (i.e., 15 and 20% BM and 9 and 8 kg) supporting the significant reduction in FT for all the speeds analyzed here. The clear difference between endurance (i.e., trail runners) runners and sprinters and the difference in additional loads (i.e., +5 and +10% BM vs. 15 and 20% BM vs. 9 and 18 kg) underpinned the differences in values between the aforementioned studies and the one here reported. Then, in contrast to a previous study where no significant changes were reported for SF when comparing unloading running with running carrying different loads (i.e., 9- and 18-kg vests) [17], a significant reduction in SF was found in the present study when running at 8 km·h^−1^ when comparing unloading running and +10% BM loaded running. Here, the difference in testing speeds (i.e., maximal velocity vs. 8 km/h) may explain such a discrepancy. When analyzing LSS, negligible to no differences were observed when comparing unloaded and loaded running with +10% BM. Silder et al. found that LSS increased significantly when comparing running with no load and 20 and 30% BM running at 12 km·h^−1^ [2]. Discrepancies in findings between both studies might be attributed to the different treadmills used (i.e., motorized treadmill vs. split-belt force-instrumented treadmill) and/or different percentages of BM (i.e., 0 vs. +10% BM and 0 vs. +20 and +30% BM).

Although a significant reduction in VO is found when comparing 0 and +10% BM, it is reasonable to think that this reduction in VO is not associated with an enhanced running economy as stated before [10] but with an inability to produce greater oscillations due to the additional load. Despite absolute MPO linearly increased in +5 and +10% BM loaded conditions (ES: 2.7 to 7.0), no differences were observed when normalized to w·kg^−1^. Although there are no previous studies assessing running power when loaded running, previous research reported that an external load of +10% BM decreased long-distance running time attributing such reduction to an increase in energy cost of running at submaximal speeds [19] and to the adoption of a running strategy that produces such an increase in running metabolic cost [2]. Such reductions in running performance have also been reported showing significant differences between unloaded and loaded (i.e., +8, +15, +20% BM) running performance, being running performance reduced by 6.9–9.9% after the application of loads [20]. As the relationship between running power and speed is highly linear (r = 0.999) when treadmill running [21], it seems logical to argue that running with an additional load of +10% BM jeopardizes running performance at submaximal speed given the increase in DF (ES: 0.7 to 1.1) here observed.

Regarding running with an additional load of +5% BM, when analyzing GCT, a significant reduction and a significant increase were observed for 8 and 10 km·h^−1^, respectively. This is partly supported by a previous study where the authors reported a significant increase in GCT when running at 12 km·h^−1^ [3]. Of note, we did not observe a significant difference between running with no load and loaded running with a +5% BM at 12 km·h^−1^ or 14 km·h^−1^. The discrepancy between values might be attributed to the different methods implemented. While we used the Stryd running powermeter for data acquisition during treadmill running, the aforementioned study completed an over-ground protocol using 3D motion cameras and force plates [3]. FT decreased significantly when assessing the differences between unloaded and +5% BM loaded running at 10 and 12 km·h^−1^. As discussed above, previous studies in sprint running reported significant reductions in FT when loaded running [17,18]. Unfortunately, and to the best of the authors’ knowledge, no study had reported yet the influence of additional load on distance runners on FT, which allows us to provide new insights into the behavior of such parameters under different conditions. As expected, after analyzing the results regarding DF, a significant increase in DF was observed when running at 10 km·h^−1^ with an additional load of +5% BM given the significant increase in GCT and reduction in FT at such velocity. SF significantly increased when running with +5% BM. This is partly supported by Silder and colleagues as they also found that FT significantly increases when running with an additional load of +5% BM [2]. Although they found significant differences at treadmill running paces ranging from 9.6 to 13.1 km·h^−1^, we only found such significance at 8 and 12 km·h^−1^. An additional study on sprint running on a treadmill also determined the influence of a load of +5% BM on step frequency finding no significant influence [22], which seems to be contradictory to our findings. Of note, endurance runners and sprinters are completely different from physiological and biomechanical standpoints. When evaluating LSS, the +5% BM additional load seems to produce enough stimuli to observe a significant increase in this key neuromuscular parameter for storage and recoil elastic energy. For all the speeds here assessed, LSS resulted to be significantly higher when loaded running than when running with no load. This might incorporate a new practical application for runners and coaches aiming to optimize LSS allowing the training to occur in their natural environment.

When analyzing MPO and nMPO, a significant increase in running power was found for both +5 and +10% BM in absolute terms, but not when normalized to BM. Of note, it seems that running with an additional load of +5% BM at 12 km·h^−1^ stimulates the production of running power, which highly correlates with speed [21] and, therefore, running performance. However, while the increase in MPO in the +5% BM condition aligns with the LSS increase (i.e., without altering the rest of the kinematic parameters), the contribution of the elastic component does not seem to be as helpful in the +10% BM condition, forcing massive modification of the rest of running parameters to ensure power production.

A previous study on the acute effects of weighted vest training [23] showed significant improvement in maximal running speed (2.9 ± 0.8%) and running economy (6.0 ± 1.6%), with small–moderate reductions in cardiorespiratory measures (ICC 90%). Furthermore, the authors stated that changes in leg stiffness could explain these performance improvements. In this sense, the results of the present work represent a step forward in the specificity of trail running training as they reveal a dose–response relationship between the magnitude of the extra load and the elastic behavior of the leg spring.

Despite the findings, we acknowledge that there are several study limitations that require consideration. First, only trained male trail runners were included in the recruitment remaining thus unknown how the findings showed in the present study may behave for either recreational, elite, or female athletes. All the measurements were completed indoors on a level treadmill leaving the outcomes for overground running unknown. However, a recent review stated that treadmill running is largely comparable to overground running; particularly, spatiotemporal parameters are related parameters [24], as the ones reported here. Future research is required into the analysis of the likely metabolic and mechanical long-term adaptations of a training intervention using additional loads on the parameters here analyzed and its influence on long-distance running performance.

## 5. Conclusions

Running with an additional load seems to have the potential to improve running performance. However, sport practitioners must be cautious when selecting the appropriate amount of additional load to use. As reported in this study, an additional load equivalent to +10% BM may negatively affect running performance at the submaximal speeds evaluated, leading to reduced LSS, longer GCT, and higher DF, to ensure power generation. In contrast, an additional load equivalent to +5% BM appears to provide enough stimuli to generate more running power and enhance LSS without affecting GCT, DF, SF, or VO resulting, therefore, in improved running performance.

## Figures and Tables

**Figure 1 sensors-23-06411-f001:**
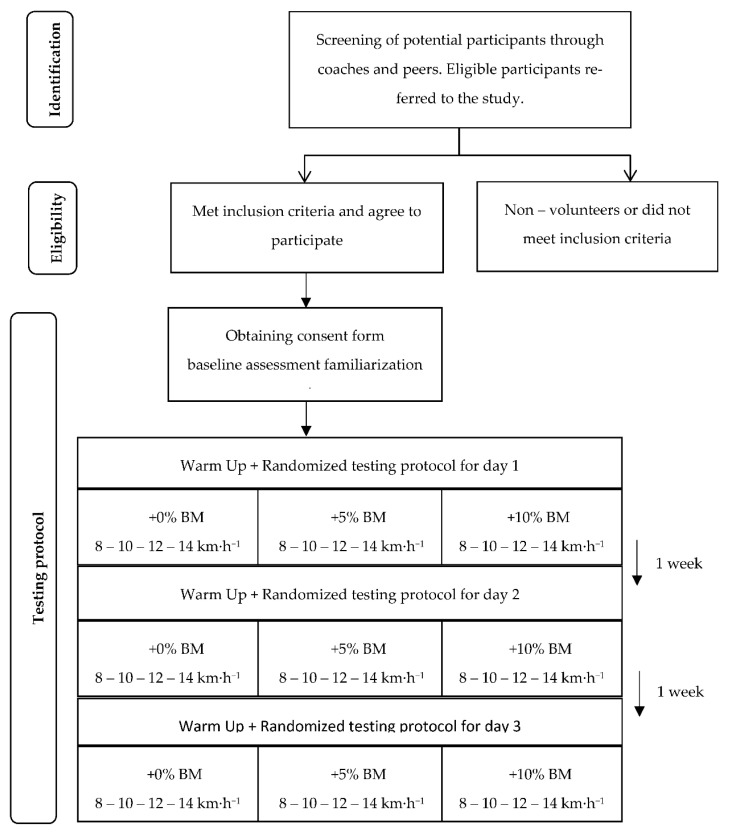
Project design timeline.

**Figure 2 sensors-23-06411-f002:**
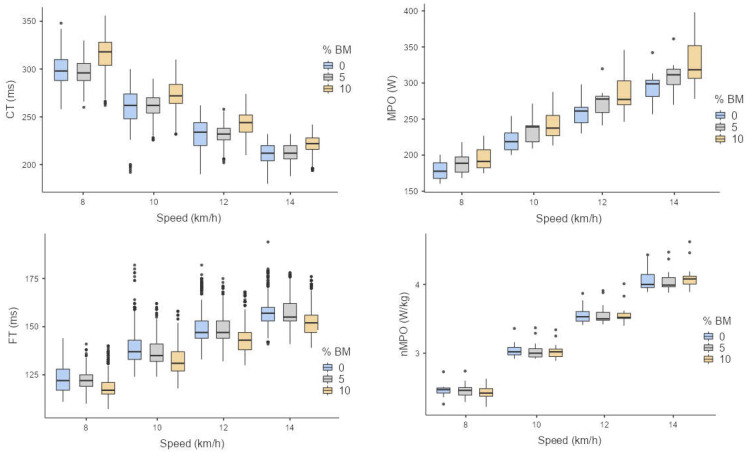
Box plots of the acute response of running power and kinematic variables to the +0%, +5%, and +10% body mass conditions at 8, 10, 12, and 14 km·h^−1^. Dots represent outliers.

**Table 1 sensors-23-06411-t001:** Descriptive characteristics of the participants (mean (SD)).

Variable	Mean (SD)
Age (years)	37.4 (5.8)
Height (m)	176.2 (4.5)
Weight (kg)	72.6 (4.9)
BMI (m/kg^2^)	23.4 (1.9)
Body fat %	11.9 (3.4)
Km/week	49.7 (20.7)

BMI: body mass index; Km/week: Run kilometers per week.

**Table 2 sensors-23-06411-t002:** Acute response of running power and kinematic variables (mean ± SD) to the different overweighted conditions and speeds. The main differences ± SD between conditions are also shown along with Cohen’s d for effect size.

Speed	Variable	+0% BM	+5% BM	+10% BM	0 vs. 5% BM	ES	0 vs. 10% BM	ES	5 vs. 10% BM	ES
**8 km·h^−1^**	Power (w)	179 ± 3	187 ± 4	195 ± 4	−9 ± 5	2.7	−16 ± 5 *	5.3	−7 ± 5	2.0
	Power (w/kg)	2.46 ± 0.03	2.47 ± 0.03	2.43 ± 0.03	−0.01 ± 0.03	0.3	0.02 ± 0.04	1.0	0.05 ± 0.01	1.3
	LSS (kN/m)	11.36 ± 1.17	11.71 ± 1.22	11.34 ± 1.52	−0.35 ± 0.05 *	0.3	0.02 ± 0.06	0.0	0.37 ± 0.06	0.0
	GCT (ms)	299 ± 17	297 ± 12	316 ± 19	2 ± 1 *	0.2	−17 ± 1 *	0.8	−20 ± 1 *	0.9
	FT (ms)	123 ± 6	123 ± 5	118 ± 5	0.1 ± 0.2	0.0	4.3 ± 0.2 *	0.8	4.3 ± 0.2 *	0.9
	DF (%)	35.4 ± 1.0	35.4 ± 0.7	36.4 ± 0.9	0.1 ± 0.0	0.0	−0.9 ± 0.1 *	0.9	−1.0 ± 0.1 *	1.1
	SF (spm)	164 ± 5	165 ± 6	161 ± 7	−1 ± 0.3 *	0.2	3 ± 0.3 *	0.5	4 ± 0.3 *	0.7
	SL (cm)	81 ± 4	81 ± 4	81 ± 4	0 ± 0	0.0	−1 ± 0 *	0.2	−1 ± 0 *	0.2
	VO (cm)	6.6 ± 0.7	6.5 ± 0.7	6.4 ± 0.7	0.1 ± 0.0	0.1	0.1 ± 0.0 *	0.2	0.1 ± 0.0	0.1
**10 km·h^−1^**	Power (w)	221 ± 4	232 ± 4	243 ± 5	−11 ± 7	2.7	−22 ± 7 *	5.5	−11 ± 7	2.7
	Power (w/kg)	3.04 ± 0.03	3.04 ± 0.04	3.03 ± 0.03	0.00 ± 0.01	0.0	0.01 ± 0.05	0.3	0.00 ± 0.05	0.3
	LSS (kN/m)	11.57 ± 1.28	11.89 ± 1.48	11.65 ± 1.49	−0.32 ± 0.07 *	0.2	−0.08 ± 0.07	0.1	0.24 ± 0.07 *	0.2
	GCT (ms)	259 ± 15	262 ± 12	271 ± 16	−3 ± 1 *	0.2	−12 ± 1 *	0.8	−10 ± 1 *	0.7
	FT (ms)	139 ± 8	137 ± 6	133 ± 8	2.1 ± 0.3 *	0.3	5.9 ± 0.4 *	0.7	3.8 ± 0.3 *	0.5
	DF (%)	32.5 ± 1.2	32.8 ± 0.9	33.5 ± 1.2	−0.3 ± 0.1 *	0.3	−1.0 ± 0.1 *	0.8	−0.7 ± 0.1 *	0.7
	SF (spm)	167 ± 6	168 ± 7	167 ± 7	−1 ± 0.3	0.1	0.4 ± 0.3	0.1	1 ± 0.3 *	0.2
	SL (cm)	100 ± 5	100 ± 5	100 ± 5	0 ± 0	0.0	1 ± 0	0.0	−1 ± 0	0.0
	VO (cm)	7.4 ± 0.8	7.3 ± 0.8	7.1 ± 0.9	0.1 ± 0.0 *	0.2	0.3 ± 0.0 *	0.4	0.2 ± 0.0 *	0.2
**12 km·h^−1^**	Power (w)	258 ± 4	272 ± 5	286 ± 7	−14 ± 8	3.5	−28 ± 8 *	7.0	−14 ± 8	2.8
	Power (w/kg)	3.56 ± 0.03	3.57 ± 0.04	3.57 ± 0.04	−0.01 ± 0.05	0.3	−0.02 ± 0.05	0.3	0.01 ± 0.05	0.0
	LSS (kN/m)	11.78 ± 1.38	12.25 ± 1.41	11.74 ± 1.33	−0.47 ± 0.07 *	0.3	0.04 ± 0.07 *	0.1	0.51 ± 0.07 *	0.4
	GCT (ms)	233 ± 23	233 ± 23	242 ± 24	0 ± 1	0.1	−10 ± 1 *	0.4	−10 ± 0.5 *	0.4
	FT (ms)	149 ± 9	148 ± 6	144 ± 8	0.9 ± 0.3 *	0.1	5.4 ± 0.4 *	0.7	4.5 ± 0.3 *	0.7
	DF (%)	30.4 ± 1.3	30.5 ± 0.7	31.4 ± 1.1	−0.1 ± 0.1	0.1	−0.9 ± 0.1 *	0.8	−0.8 ± 0.1 *	1.0
	SF (spm)	174 ± 6	174 ± 8	173 ± 7	−0.8 ± 0.3 *	0.1	0.6 ± 0.3	0.0	1.4 ± 0.3 *	0.1
	SL (cm)	116 ± 5	116 ± 6	116 ± 5	−0 ± 0	0.0	−0 ± 0	0.1	−0 ± 0	0.1
	VO (cm)	7.4 ± 0.8	7.4 ± 0.9	7.2 ± 0.9	0.0 ± 0.0	0.0	0.2 ± 0.0 *	0.2	0.2 ± 0.0 *	0.2
**14 km·h^−1^**	Power (w)	295 ± 5	309 ± 5	329 ± 8	−15 ± 9	2.8	−34 ± 9 *	6.8	−19 ± 9 *	4.0
	Power (w/kg)	4.06 ± 0.04	4.06 ± 0.04	4.11 ± 0.05	−0.00 ± 0.06	0.0	−0.05 ± 0.06	1.3	0.04 ± 0.06	1.2
	LSS (kN/m)	11.78 ± 1.33	12.24 ± 1.52	11.66 ± 1.27	−0.47 ± 0.07 *	0.2	0.12 ± 0.07	0.1	0.59 ± 0.07 *	0.4
	GCT (ms)	212 ± 21	212 ± 21	221 ± 22	0 ± 0	0.0	−9 ± 1 *	0.4	−9 ± 1 *	0.4
	FT (ms)	158 ± 7	157 ± 7	152 ± 9	0.3 ± 0.3	0.0	5.9 ± 0.4 *	0.7	5.6 ± 0.4 *	0.7
	DF (%)	28.7 ± 1.0	28.7 ± 0.9	29.6 ± 1.1	−0.0 ± 0.0	0.0	−0.9 ± 0.1 *	0.9	−0.9 ± 0.1 *	0.9
	SF (spm)	180 ± 7	180 ± 8	180 ± 9	−0.3 ± 0.4	0.0	0.2 ± 0.4	0.0	0.5 ± 0.4	0.1
	SL (cm)	129 ± 5	130 ± 6	130 ± 7	−1 ± 0	0.1	−1 ± 0	0.1	−0 ± 0	0.0
	VO (cm)	7.3 ± 0.8	7.3 ± 0.8	7.1 ± 1.0	0.0 ± 0.0	0.0	0.2 ± 0.0 *	0.3	0.2 ± 0.0 *	0.3

* *p* < 0.05; ES: effect size; BM: body mass; +5% or +10% BM: running with a 5% or 10% of body mass in the weighted vest; LSS: leg spring stiffness; GCT: ground contact time; FT: flight time; DF: duty factor; SF: step frequency; SL: step length; VO: vertical oscillation.

## Data Availability

The data presented in this study are available on request from the corresponding author. The data are not publicly available due to authors’ preferences.

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
