# Peer review of "Training Specificity in Trail Running: A Single-Arm Trial on the Influence of Weighted Vest on Power and Kinematics in Trained Trail Runners"

_sensors, 2023, doi:10.3390/s23146411_

Round 1
Reviewer 1 Report
1) According to figure 2, this conclusion “running with an additional load of 5% body mass produced an increase in running LSS” can not be naturally induced.
2) Why can conclude that “ it has been confirmed that running with lighter loads (i.e., 5% BM) takes the principle of specificity in trail running one step further, enhancing running power generation and LSS.”?
3) There is no cues to indicate that “at 12 km/h a significant reduction in LSS was also found when running with +10% BM”
4) As the author says ”When analyzing LSS, it was found that a significant reduction was observed when running at 12 km/h when comparing unloaded and loaded running with +10%BM.” Why can safely give this conclusion? The significance analysis is needed.
5) What’s physical meaning of the running performance, why the running performance can be revealed by the running power and leg spring stiffness?
The English language is basically well and the presented description can be well followed.
Reviewer 2 Report
Strengths: There is an advantage in designing an exercise program by suggesting additional weight for runners.
Weakness: It can be a more advanced study if conducted as a two-group study.
1. Please indicate "Single-Arm Trials" in the title.
2. Can you do a sample size calculation?
3. Were they evaluated by raters blinded to the study?
4. Please fill out the reliability and validity of the evaluation tool.
5. Please explain the abbreviations in the table.
6. Please add more clinical significance to the discussion of this article.
7. Please check the spelling and grammar of the text
Round 2
Reviewer 1 Report
The reviewer's questions are well answered. The reviewer has no other questions.
The quality of the English Language is basically well.